# A GREAT Architecture for Edge-Based Graph Problems Like TSP

## Abstract

In the last years, many neural network-based approaches have been proposed to tackle combinatorial optimization problems such as routing problems. Many of these approaches are based on graph neural networks (GNNs) or related transformers, operating on the Euclidean coordinates representing the routing problems. However, GNNs are inherently not well suited to operate on dense graphs, such as in routing problems. Furthermore, models operating on Euclidean coordinates cannot be applied to non-Euclidean versions of routing problems that are often found in real-world settings. To overcome these limitations, we propose a novel GNN-related edge-based neural model called Graph Edge Attention Network (GREAT). We evaluate the performance of GREAT in the edge-classification task to predict optimal edges in the Traveling Salesman Problem (TSP). We can use such a trained GREAT model to produce sparse TSP graph instances, keeping only the edges GREAT finds promising. Compared to other, non-learning-based methods to sparsify TSP graphs, GREAT can produce very sparse graphs while keeping most of the optimal edges. Furthermore, we build a reinforcement learning-based GREAT framework which we apply to Euclidean and non-Euclidean asymmetric TSP. This framework achieves state-of-the-art results.

## 1 Introduction

Graph neural networks (GNNs) have emerged as a powerful tool for learning on graph-structured data such as molecules, social networks, or citation graphs (Wu et al., 2020). In recent years, GNNs have also been applied in the setting of combinatorial optimization, especially routing problems (Joshi et al., 2019; Hudson et al., 2021; Xin et al., 2021) since such problems can be interpreted as graph problems. However, the graph representations of routing problems, which are typically complete, dense graphs, are ill-suited for GNNs. This is because vanilla GNNs are not generally suitable for learning on complete graphs. GNNs are related to the Weisfeiler Leman algorithm which is known to exploit graph structure (Morris et al., 2019). Complete graphs feature no such structure, resulting in poor GNN performance. Moreover, over-smoothing is a well-known problem happening in (deep) GNNs which means that feature vectors computed for different nodes become more and more similar with every layer (Rusch et al., 2023). Naturally, in dense or even complete graphs this problem is even more present as all nodes share the same information leading to similar encodings. Consequently, Lischka et al. (2024) showed that the performance of GNNs operating on routing problems can be increased if graphs are made sparse in a preprocessing step. However, the proposed sparsification methods of Lischka et al. (2024) rely on hand-crafted heuristics which goes against the idea of data-driven, end-to-end machine learning frameworks.

In this paper, we overcome the limitations of regular GNNs by introducing the Graph Edge Attention Network (GREAT). This results in the following contributions:

- Whereas traditional GNNs operate on a node-level by using node-based message passing operations, GREAT is edge-based, meaning information is passed along edges sharing endpoints. This makes GREAT perfect for edge-level tasks such as routing problems where the edges to travel along are selected. We note, however, that the idea of GREAT is task-independent and it can potentially also be applied in other suitable settings, possibly chemistry or road networks.

- We evaluate GREAT in the task of edge classification, training the architecture to predict optimal edges in a Traveling Salesman Problem (TSP) tour in a supervised setting. By this, GREAT can be used as a learning-based and data-driven sparsification method for routing graphs. The produced sparse graphs are less likely to delete optimal edges than hand-crafted heuristics while being overall sparser.

- We build a reinforcement learning framework that can be trained end-to-end to predict optimal TSP tours. As the inputs of GREAT are edge features (e.g., distances), GREAT applies to all variants of TSP, including non-Euclidean variants such as the asymmetric TSP. The resulting trained framework achieves state-of-the-art performance for two asymmetric TSP distributions.

## 2 BASICS AND RELATED WORK

### 2.1 GRAPH NEURAL NETWORKS

Graph neural networks are a class of neural architectures that operate on graph-structured data. In contrast to other neural architectures like MLPs where the connections of the neurons are fixed and grid-shaped, the connections in a GNN reflect the structure of the input data.

In essence, GNNs are a neural version of the well-known Weisfeiler-Leman (WL) graph isomorphism heuristic (Morris et al., 2019; Xu et al., 2019). In this heuristic, graph nodes are assigned colors that are iteratively updated. Two nodes share the same color if they shared the same color in the previous iteration and they had the same amount of neighbors of each color in the last iteration. When the test is applied to two graphs and the graphs do not have the same amount of nodes of some color in some iteration, they are non-isomorphic. WL is only a heuristic, however, as there are certain non-isomorphic graphs it can not distinguish. Examples are regular graphs (graphs where all nodes have the same degree) (Kiefer, 2020). We note that complete graphs (that we encounter in routing problems) are regular graphs.

GNNs follow a similar principle as the WL heuristic, but instead of colors, vector representations of the nodes are computed. GNNs iteratively compute these node feature vectors by aggregating over the node feature vectors of adjacent nodes and mapping the old feature vector together with the aggregation to a new node feature vector. Additionally, the feature vectors are multiplied with trainable weight matrices and non-linearities are applied to achieve actual learning. The node feature vectors of a neighborhood are typically scaled in some way (depending on the respective GNN architecture) and sometimes, edge feature vectors are also considered within the aggregations. An example can be found in fig. 1. Its mathematical formulation might look like this:

$$h_v^i = \sigma\big(W_1^i h_v^{i-1} + \sum_{u \in N(v)} (W_2^i h_u^{i-1} + W_3^i e_{vu})\big) \tag{1}$$

here $W_1^i, W_2^i, W_3^i$ are trainable weight matrices of suitable sizes, $\sigma$ is a non-linear activation function, $h_u^i$ denotes the feature vector of a node $u$ in the $i$th update of the GNN and $e_{uv}$ denotes an edge feature of the edge $(u, v)$ in the input graph. Sometimes, the edge features are also updated. The node feature vectors of the last layer of the GNN can be used for node-level classification or regression tasks. They can also be summarized (e.g. by aggregation) and used as a graph representation in graph-level tasks. Referring back to the WL algorithm, we note how the node colors there can also be considered as node classes. Furthermore, comparing the colors of different graphs to determine isomorphism can be considered a graph-level task. While GNNs are bounded in their expressiveness by the WL algorithm (Morris et al., 2019; Xu et al., 2019) and can therefore not distinguish regular graphs (e.g., complete graphs), we acknowledge that these limitations of GNNs can be mitigated by assigning unique "node identifiers" (like unique node coordinates) to graphs passed to GNNs (Abboud et al., 2021). However, over-smoothing (Rusch et al., 2023) is still a problem, especially in dense graphs. This results in a need for better neural encoder architectures in settings such as the routing problem.

## 2.2 ATTENTION-BASED GRAPH NEURAL NETWORKS

Graph Attention Networks (GATs (Velickovic et al., 2017)) are a variety of GNNs. They leverage the attention mechanism (Vaswani et al., 2017) to determine how to scale the messages sent between the network nodes. Overall, the node features are computed as follows:

$$x'_i = \sum_{j \in N(i) \cup \{i\}} \alpha_{i,j} \Theta_t x_j \tag{2}$$

where $\alpha_{i,j}$ is computed as

$$\alpha_{i,j} = \frac{\exp(\text{LeakyReLU}(\mathbf{a}_s^\top \Theta_s x_i + \mathbf{a}_t^\top \Theta_t x_j + \mathbf{a}_e^\top \Theta_e e_{i,j}))}{\sum_{k \in N(i) \cup \{i\}} \exp(\text{LeakyReLU}(\mathbf{a}_s^\top \Theta_s x_i + \mathbf{a}_t^\top \Theta_t x_k + \mathbf{a}_e^\top \Theta_e e_{i,k}))} \tag{3}$$

and $\Theta_e, \Theta_s, \Theta_t.\mathbf{a}_e^\top, \mathbf{a}_s^\top, \mathbf{a}_t^\top$ are learnable parameters. *We note that GAT uses the edge features only to compute the attention scores but does not update them nor uses them in the actual message passing.*

Variations of the original GAT also use edge features in the message-passing operations. For example, Chen & Chen (2021) propose Edge-Featured Graph Attention Networks (EGAT) which uses edge features by applying a GAT not only on the input graph itself but also its line graph representation (compare Chen et al. (2017) as well) and then combining the computed features.

Another work incorporating edge features in an attention-based GNN is Shi et al. (2020) who use a "Graph Transformer" that incorporates edge and node features for a semi-supervised classification task.

Jin et al. (2023a) introduce "EdgeFormers" an architecture operating on Textual-Edge Networks where they combine the success of Transformers in LLM tasks and GNNs. Their architecture also augments GNNs to utilize edge (text) features.

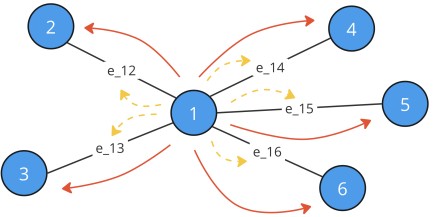

Figure 1: Classical GNN: Node attends to neighboring nodes (+ optionally to adjacent edges)

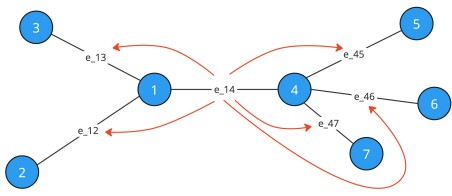

Figure 2: GREAT (node-free): Edge attends to adjacent edges

## 2.3 LEARNING TO ROUTE

In recent years, many studies have tried to solve routing problems such as the Traveling Salesmen Problem or the Capacitated Vehicle Routing Problem (CVRP).

Popular approaches for solving routing problems with the help of machine learning include reinforcement learning (RL) frameworks, where encodings for the problem instances are computed. These encodings are then used to incrementally build solutions by selecting one node in the problem instance at a time. Successful works in this category include Deudon et al. (2018); Nazari et al. (2018); Kool et al. (2019); Kwon et al. (2020); Jin et al. (2023b).

Another possibility to use machine learning for solving routing problems is to predict edge probabilities or scores which are later used in search algorithms such as beam search or guided local search. Examples for such works are Joshi et al. (2019); Fu et al. (2021); Xin et al. (2021); Hudson et al. (2021); Kool et al. (2019); Min et al. (2024).

A further possibility is iterative methods where a solution to a routing problem is improved over and over until a stopping criterion (e.g., convergence) is met. Possibilities for such improvements are optimizing subproblems or applying improvement operators such as $k$-opt. Examples for such works are da Costa et al. (2021); Wu et al. (2021); Cheng et al. (2023); Lu et al. (2019); Chen & Tian (2019); Li et al. (2021).

### 2.3.1 NON-EUCLIDEAN ROUTING PROBLEMS

Many of the mentioned works, especially in the first two categories, use GNNs or transformer models (which are related to GNNs via GATs (Joshi, 2020)) to capture the structure of the routing problem in their neural architecture. This is done by interpreting the coordinates of Euclidean routing problem instances as node features. These node features are then processed in the GNN or transformer architectures to produce encodings of the overall problem. However, this limits the applicability of such works to Euclidean routing problems. This is unfortunate, as non-Euclidean routing problems are highly relevant in reality. Consider, e.g., one-way streets which result in unequal travel distances between two points depending on the direction one is going. Another example is variants of TSP that consider energy consumption as the objective to be minimized. If point A is located at a higher altitude than point B, traveling from A to B might require less energy than the other way around.

So far, only a few studies have also investigated non-Euclidean versions of routing problems, such as the asymmetric TSP (ATSP). One such study is Gaile et al. (2022) where they solve synthetic ATSP instances with unsupervised learning, reinforcement learning, and supervised learning approaches. Another study is Wang et al. (2023) that uses online reinforcement learning to solve ATSP instances of TSPLIB (Reinelt, 1991). Another successful work tackling ATSP is Kwon et al. (2021). There, the *Matrix Encoding Network (MatNet)* is proposed, a neural model suitable to operate on matrix encodings representing combinatorial optimization problems such as the distance matrices of (A)TSP. Their model is trained using RL.

## 3 GRAPH EDGE ATTENTION NETWORK

Existing GNNs are based on node-level message-passing operations, making them perfect for node-level tasks as is also underlined by their connection to the WL heuristic. In contrast, we propose an edge-level-based GNN where information is passed along neighboring edges. This makes our model perfect for edge-level tasks such as edge classification (e.g., in the context of routing problems, determining if edges are "promising" to be part of the optimal solution or not). Our model is attention-based, meaning the "focus" of an edge to another, adjacent edge in the update operation is determined using the attention mechanism. Consequently, similar to the Graph Attention Network (GAT) we call our architecture Graph Edge Attention Network (GREAT). A simple visualization of the idea of GREAT is shown in fig. 2. In this visualization, edge $e_{14}$ attends to all other edges it shares an endpoint with. While GREAT is a task-independent framework, it is suited perfectly for routing problems: Consider TSP as an example. There, we do not have any node features, only edge features given as distances between nodes. A normal GNN would not be suitable to process such information well. Existing papers use coordinates of the nodes in the Euclidean space as node features to overcome this limitation. However, this trick only works for Euclidean TSP and not other symmetric or asymmetric TSP adaptations. GREAT, however, can be applied to all these variants.

Instead of purely focusing on edge features and ignoring node features, it would also be possible to transform the graph in its line graph and apply a GNN operating only on node features on this line graph. In the line graph, each edge $e_{i,j}$ of the original graph is a node $n_{i,j}$ and two nodes $n_{i,j}, n_{k,m}$ in the line graph are connected if the corresponding edges $e_{i,j}$ and $e_{k,m}$ in the original graph share an endpoint. However, a TSP instance of $n$ cities contains $n^2$ many edges. Therefore, the line graph of this instance would have $\mathcal{O}(n^2)$ many nodes. Furthermore, as the original TSP graph is complete, each of the endpoints $\{i, j\}$ of an edge in the original graphs is part of $n$ many edges. This means that in the line graph, each node has $2n$ many connections to other nodes. In other words, each of the $\mathcal{O}(n^2)$ nodes has $\mathcal{O}(n)$ edges leading to $\mathcal{O}(n^3)$ many edges in the line graph. This implies that the line graph has one order of magnitude more edges and nodes than the original graph.

We note that GREAT can be applied to extensions of the TSP such as the CVRP or TSP with time windows (TSPTW) as well: even though capacities and time windows are node-level features, we

can easily transform them into edge features. Consider CVRP where a node $j$ has a demand $c_j$. We can simply add demand $c_j$ to all edges $e_{i,j}$ in the graph. This is because we know that if we have an edge $e_{i,j}$ in the tour, we will visit node $j$ in the next step and therefore need a free capacity in our vehicle big enough to serve the demand of node $j$ which is $c_j$. An analogous extension works for TSPTW.

We further note that even though GREAT has been developed in the context of routing problems, it generally is a task-oblivious architecture and it might be useful in completely different domains as well such as chemistry, road, or flow networks.

## 3.1 Architecture

In the following, we provide the mathematical model defining the different layers of a GREAT model. In particular, we propose two versions of GREAT.

The first version is purely edge-focused and does not have any node features. Here, each edge exchanges information with all other edges it shares at least one endpoint with. The idea essentially corresponds to the visualization in fig. 2. In the following, we refer to this variant as "node-free" GREAT.

The second version is also edge-focused but has intermediate, temporary node features. This essentially means that nodes are used to save all information on adjacent edges. Afterward, the features of an edge are computed by combining the temporary node features of their respective endpoints. These node features are then deleted and *not* passed on to the next layer, only the edge features are passed on. The idea of this GREAT variant is visualized in fig. 3 and fig. 4 In the remainder of this study, we refer to this GREAT version as "node-based".

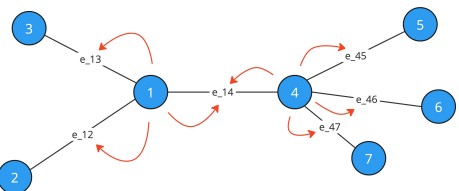

Figure 3: GREAT (node-based): compute temporary node features

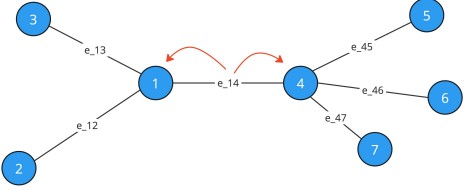

Figure 4: GREAT (node-based): combine temporary node features

## 3.2 Mathematical Formulations

We now describe the mathematical formulas defining the internal operations of GREAT. We note that, inspired by the original transformer architecture of Vaswani et al. (2017), GREAT consists of two types of sublayers: attention sublayers and feedforward sublayers. We always alternate between attention and feedforward sublayers. The attention sublayers can be node-based (with temporary nodes features) or completely node-free. Using the respective sublayers leads to overall node-based or node-free GREAT. A visualization of the architecture can be found in fig. 5.

**Node-Based GREAT, Attention Sublayers**: For each node in the graph, we compute a temporary node feature

$$x_i = \sum_{j \in N(i)} (\alpha'_{i,j} W'_1 e_{i,j} || \alpha''_{i,j} W''_1 e_{j,i}) \tag{4}$$

with

$$\alpha'_{i,j} = \text{softmax}\big(\frac{(W'_2 e_{i,j})^\top W'_3 e_{i,j}}{\sqrt{d}}\big), \alpha''_{i,j} = \text{softmax}\big(\frac{(W''_2 e_{j,i})^\top W''_3 e_{j,i}}{\sqrt{d}}\big) \tag{5}$$

Note that we compute two attention scores and concatenate the resulting values to form the temporary node feature. This allows GREAT to differentiate between incoming and outgoing edges which, e.g. in the case of asymmetric TSP, can have different values. If symmetric graphs are processed (where $e_{i,j} = e_{j,i}$ for all nodes $i, j$) we can simplify the expression to only one attention score.

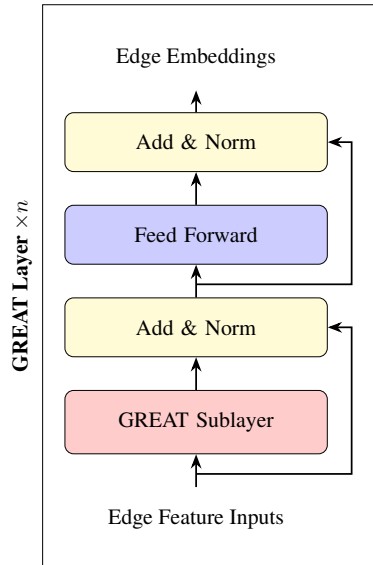

Figure 5: A GREAT layer with sublayers and normalizations

The temporary node features are concatenated and mapped to the hidden dimension again to compute the actual edge features of the layer.

$$e'_{i,j} = W_4(x_i||x_j) \tag{6}$$

We note that $W'_1, W''_1, W'_2, W''_2, W'_3, W''_3, W'_4, W''_4$ are trainable weight matrices of suitable dimension. $d$ is the hidden dimension and $||$ denotes concatenation. $W'_1 e_{i,j}, W'_2 e_{i,j}$ and $W'_3 e_{i,j}$ correspond to the "values", "keys" and "queries" of the original transformer architecture.

**Node-Free GREAT, Attention Sublayers**: Here, edge features are computed directly as

$$e_{i,j} = (\alpha'_{i,j}W'_1 e_{i,j}||\alpha'_{j,i}W'_1 e_{j,i}||\alpha''_{i,j}W''_1 e_{i,j}||\alpha''_{j,i}W''_1 e_{j,i}) \tag{7}$$

Note that the edge feature consists of four individual terms that are concatenated. Due to the attention mechanism, these terms summarize information on all edges outgoing from node $i$, ingoing to node $i$, outgoing from node $j$, and ingoing to node $j$. The differentiation between in- and outgoing edges is again necessary due to asymmetric graphs. The $\alpha'$ and $\alpha''$ scores are computed as for the node-based GREAT variant.

**Feedforward (FF) Subayer**: Like in the original transformer architecture, the FF layer has the following form.

$$e'_{i,j} = W_2\text{ReLU}(W_1 e_{i,j} + b_1) + b_2 \tag{8}$$

where $W_1, b_1, W_2, b_2$ are trainable weight matrices and biases of suitable sizes. Moreover, again like in Vaswani et al. (2017), the feedforward sublayers have internal up-projections, which temporarily double the hidden dimension before scaling it down to the original size.

We further note that we add residual layers and normalizations to each sublayer (Attention and FF). Therefore the output of each sublayer is (like in the original transformer architecture):

$$e'_{i,j} = \text{LayerNorm}(e_{i,j} + \text{Sublayer}(e_{i,j})) \tag{9}$$

## 4 EXPERIMENTS

We evaluate the performance of GREAT in two types of experiments. First, we train GREAT in a supervised fashion to predict optimal TSP edges. Secondly, we train GREAT in a reinforcement learning framework to construct TSP solutions incrementally directly. Our code was implemented in Python using PyTorch (Paszke et al., 2019) and Pytorch Geometric (Fey & Lenssen, 2019). The code for our experiments, trained models, and test datasets will be publicly available after the paper is accepted.

Table 1: Precision and recall determining optimal TSP edges

| p or k | Precision | Recall | # edges |
|---|---|---|---|
| **GREAT node-free** | | | |
| p = 0.00001 | 38.39% | 99.95% | 52 077 |
| p = 0.5 | 54.95% | 99.4% | 36 179 |
| p = 0.999 | 64.32% | 98.06% | 30 491 |
| **GREAT node-based** | | | |
| p = 0.00001 | 36.22% | 99.95% | 55 196 |
| p = 0.5 | 59.86% | 99.38% | 33 203 |
| p = 0.999 | 70.66% | 97.48% | 27 594 |
| **1-Tree** | | | |
| k = 10 | 19.99% | 99.97% | 100 000 |
| k = 5 | 39.65 % | 99.13% | 50 000 |
| k = 3 | 63.79% | 95.69% | 30 000 |
| **k-nn** | | | |
| k = 10 | 19.84% | 99.22% | 100 000 |
| k = 5 | 37.34 % | 93.34% | 50 000 |
| k = 3 | 54.66% | 81.98% | 30 000 |

## 4.1 LEARNING TO SPARSIFY

In this experiment, we demonstrate GREAT's capability in edge-classification tasks. In particular, we train the network to predict optimal TSP edges for Euclidean TSP of size 100. The predicted edges obtained from this network could later on be used in beam searches to create valid TSP solutions, or, alternately for TSP sparsification as done in Lischka et al. (2024). Therefore, we will evaluate the capability of the trained network for sparsifying TSP graphs and, while doing so, keeping optimal edges.

The hyperparameters in this setting are as follows. We train a node-based and a node-free version of GREAT. For both models, we choose 10 hidden layers. The hidden dimension is 64 and each attention layer has 4 attention heads. Training is performed for 200 epochs and there are 50,000 training instances in each epoch. Every 10 epochs, we change the dataset to a fresh set of 50,000 instances (meaning $200 \times 50,000 = 10,000,000$ instances in total). We used the Adam optimizer with a constant learning rate of 0.0001 and weighted cross-entropy as our loss function to account for the fact that there is an unequal number of optimal and non-optimal edges in a TSP graph. Targets for the optimal edges were generated using the LKH algorithm.

The evaluation of the trained networks is done on 100 instances. The performance of the network is benchmarked against the results of the "classical" algorithms 1-Tree and k-nn used in the graph sparsification task of Lischka et al. (2024). The results are shown in table 1. For the "classical" sparsification methods, we can set a hyperparameter $k$ specifying that the $k$ most promising outgoing edges of each node in the graph are kept in the sparsified instance. This parameter $k$ allows us to perfectly influence how many edges will be part of the sparse graph ($k \times n$, where $n$ is the TSP size). We chose three different values of $k$, i.e., $k = 3, 5, 10$. For GREAT, there is no such a hyperparameter. We can, instead, set different thresholds for the probability $p$ that the network predicts an edge to be part of the optimal solution. For this, we chose $0.00001, 0.5$, and $0.999$. We can see that choosing $p = 0.999$ results in a similar number of edges as $k = 3$. In this setting, the precision and recall of both GREAT versions are significantly better than the scores achieved by the "classical" algorithms. Here, by precision, we quantify the performance of only keeping edges that are indeed optimal. Recall refers to the ability of the approach to keep all optimal edges in the sparse graph. The result indicates that GREAT can produce very sparse graphs while missing relatively few optimal edges. Overall, however, we can see that for the classical algorithms, it is easier to just make the graphs less sparse and by this prevent deleting optimal edges. For GREAT, this is not possible, as lowering $p$ further to increase recall leads to prohibitively low precision.

Overall, we summarize that GREAT is a very powerful technique for creating extremely sparse TSP graphs while deleting only a small number of optimal edges. We hypothesize that creating such

very sparse but "optimal" graphs can be beneficial for the ensemble methods of Lischka et al. (2024) where sparse graphs of the most promising edges are combined with dense graphs to prevent deleting optimal edges completely. We further observe that node-free and node-based GREAT achieve rather similar results in this experiment.

## 4.2 Learning to Solve non-Euclidean TSP

In this task, we train GREAT in a reinforcement learning framework to construct TSP solutions incrementally by adding one node at a time to a partial solution. Our framework follows the encoder-decoder approach (where GREAT serves as the encoder and a multi-pointer network as the decoder) and is trained using POMO (Kwon et al., 2020). We focus on three different TSP variants, and by this aim to demonstrate GREAT's versatility to also apply to non-Euclidean TSP:

1. Euclidean TSP where the coordinates are distributed uniformly at random in the unit square.
2. Asymmetric TSP with triangle inequality (TMAT) as was used in Kwon et al. (2021). We use the code of Kwon et al. (2021) to generate instances. However, we normalize the distance matrices differently: Instead of a fixed scaling value, we normalize each instance individually such that the biggest distance is exactly 1. By this, we ensure that the distances use the full range of the interval (0,1) as well as possible.
3. Extremely asymmetric TSP (XASY) where all pairwise distances are sampled uniformly at random from the interval (0,1). The same distribution was used in Gaile et al. (2022). Here, the triangle inequality does generally not hold.

The exact setting in this experiment is the following. For each distribution, we train three versions of GREAT. A node-based and a node-free network with hidden dimension 128 as well as a node-free network with hidden dimension 256. All networks have 5 hidden layers and 8 attention heads. Training is done for 400 epochs and there are 25,000 instances in each epoch. Again, every 10 epochs, we change the dataset to a fresh set of 25,000 instances (meaning $400 \times 25,000 = 10,000,000$ instances in total). We evaluate the model after each epoch and save the model with the best validation loss during these 400 epochs for testing. Furthermore, while training, the distances of all instances in the current data batch were multiplied by a factor in the range (0.5, 1.5) to ensure the models learn from a more robust data distribution. This allows us to augment the dataset at inference by a factor of $\times 8$ like was done in Kwon et al. (2020). However, we want to note that while augmenting the data by this factor at inference time improves performance, using even bigger augmentation factors like $\times 128$ in Kwon et al. (2021) does not lead to much better results (especially considering the enormous blowup in runtime). We suppose that this is due to our augmentation implementation having a disadvantage. While the augmentation method in, e.g., Kwon et al. (2020) which works by rotating coordinates, does not alter the underlying distribution much, our method by multiplying distances does change the distribution considerably. We note that instances multiplied with values close to 1 are favored in the end, compared to instances multiplied with values close to the borders 0.5 and 1.5.

The overall framework to construct solutions, as well as the decoder to decode the encodings provided by GREAT and the loss formulation, are adapted from Jin et al. (2023b). We note that in this setting of incrementally building TSP solutions with an encoder-decoder approach, we would like to have *node encodings* as input for the decoder and not *edge encodings* like they are produced by GREAT. This is because we want to iteratively select the next node to visit, given a partial solution. As GREAT is generally used to compute edge encodings, all GREAT architectures in this experiment (node-free and node-based) have a final node-based layer where the results of the temporary node features (compare fig. 3) are returned instead of processing them further to obtain edge embeddings again. By this, we can provide the decoder architecture with node encodings, despite having operated on edge-level during the internal operations of GREAT. A visualization of the framework can be found in fig. 6.

In the following, we provide an overview of the performance of our models in table 2 table 3 and table 4. Optimality gaps of our approaches are computed w.r.t. the optimal solver Gurobi (Gurobi Optimization, LLC, 2024). These (average) optimality gaps indicate how much worse the found solutions are in percent compared to the optimal solutions. When interpreting these results, we also point out the significant differences in the number of model parameters and the number of training instances.

For Euclidean TSP, we observe that our model does not quite achieve the performance of existing architectures. However, we note that MatNet, which operates on distance matrices, also seems to struggle, compared to the models operating on coordinates like the attention model (AM) with POMO. MatNet still performs somewhat better than GREAT, however, we attribute this mainly to the fact that MatNet has more parameters and, moreover, has been trained on a dataset more than 10 times larger than ours. Within the different GREAT architectures, the node-free versions perform better than the node-based model. Furthermore, the GREAT with the most parameters, performs best.

The TMAT distribution has several differences compared to the Euclidean distribution. Simple heuristics like nearest insertion (NI), farthest insertion (FI), and nearest neighbor (NN) perform considerably worse compared to the Euclidean case. Moreover, on TMAT, the only other available neural solver is MatNet. We note that for MatNet different distances are reported because the distances in the MatNet framework have been normalized differently (indicated with an asterisk). The optimality gaps can still be compared, however, since both models (MatNet and GREAT) have been evaluated w.r.t. an optimal solver. We see that the node-free GREAT network with 1.26M parameters achieves better performance than MatNet (which has $\sim 5$ times more parameters and is trained on a dataset over 10 times larger) when no data augmentation is performed. MatNet has also been evaluated with a $\times 128$ data augmentation which then leads to better results. However, the runtime of MatNet in this setting is considerably worse. Within the different GREAT versions, we can see that the node-free versions perform considerably better than the node-based version. However, the node-free model with only 1.26M parameters performs better than the model with 5.00M parameters.

In the extremely asymmetric distribution (XASY) case, we note that all simple heuristics (nearest insertion, farthest insertion, and nearest neighbor) perform very poorly, achieving gaps of 185% - 310%. The node-based GREAT, however, achieves gaps of $21.51\%$ (no augmentation) and $13.24\%$ ($\times 8$ augmentation). Node-free GREAT versions achieve gaps between $30\%$ and $40\%$ without augmentation, which is, contrary to the other distributions, worse than the node-based GREAT. No other neural solvers have been evaluated on this distribution with 100 nodes. However, Gaile et al. (2022) deployed a neural model on the same distribution for instances of 50 cities. A small comparison between Gaile et al. (2022) and GREAT can be found in appendix A.

Overall, we summarize that GREAT achieves state-of-the-art performance on the asymmetric TSP distributions, despite often having fewer parameters than other architectures and being trained on smaller datasets (we expect GREAT to have an even better performance when being trained on bigger datasets). On the Euclidean distribution, node-free and node-based GREAT achieve comparable performance. However, on the TMAT distribution, node-free GREAT yields better performance while node-based GREAT leads the ranking for XASY distribution.

Table 2: Euclidean TSP

| Method | Params | Train Set | EUC100 | | |
|---|---|---|---|---|---|
| | | | Len. | Gap | Time |
| Gurobi Optimization, LLC (2024) | - | - | 7.76 | - | - |
| LKH3 Helsgaun (2017) | - | - | 7.76 | 0.0% | - |
| Nearest Insertion | - | - | 9.45 | 21.8% | - |
| Farthest Insertion | - | - | 8.36 | 7.66% | - |
| Nearest Neighbor | - | - | 9.69 | 24.86% | - |
| GREAT NB x1 | 926k | 10M | 7.88 | 1.55% | 48s |
| GREAT NB x8 | 926k | 10M | 7.85 | 1.09% | 7m |
| GREAT NF x1 | 1.19M | 10M | 7.87 | 1.46% | 61s |
| GREAT NF x8 | 1.19M | 10M | 7.84 | 1.02% | 9m |
| GREAT NF x1 | 4.74M | 10M | 7.85 | 1.21% | 2m |
| GREAT NF x8 | 4.74M | 10M | 7.82 | 0.81% | 18m |
| AM + POMO x1 Kwon et al. (2020) | 1.27M | 200M | 7.80 | 0.46% | 11s |
| AM + POMO x8 Kwon et al. (2020) | 1.27M | 200M | 7.77 | 0.14% | 1m |
| MatNet x1 Kwon et al. (2021) | 5.60M | 120M | 7.83 | 0.94% | 34s |
| MatNet x8 Kwon et al. (2021) | 5.60M | 120M | 7.79 | 0.41% | 5m |

Table 3: TMAT TSP

| Method | Params | Train Set | TMAT100 | | |
|---|---|---|---|---|---|
| | | | Len. | Gap | Time |
| Gurobi Optimization, LLC (2024) | - | - | 10.69 | - | - |
| LKH3 Helsgaun (2017) | - | - | 10.69 | 0.0% | - |
| Nearest Insertion | - | - | 14.09 | 31.8% | - |
| Farthest Insertion | - | - | 13.25 | 23.92 % | - |
| Nearest Neighbor | - | - | 14.55 | 36.04% | - |
| GREAT NB x1 | 1.26M | 10M | 11.65 | 8.97% | 61s |
| GREAT NB x8 | 1.26M | 10M | 11.41 | 6.7% | 9m |
| GREAT NF x1 | 1.26M | 10M | 11.03 | 3.12% | 62s |
| GREAT NF x8 | 1.26M | 10M | 10.93 | 2.25% | 9m |
| GREAT NF x1 | 5.00M | 10M | 11.04 | 3.22% | 2m |
| GREAT NF x8 | 5.00M | 10M | 10.96 | 2.46% | 18m |
| MatNet x1 Kwon et al. (2021) | 5.60M | 120M | 1.62* | 3.24% | 34s |
| MatNet x128 Kwon et al. (2021) | 5.60M | 120M | 1.59* | 0.93% | 1h |

* used different normalization method for absolute distance

Table 4: XASY TSP

| Method | Params | Train Set | XASY100 | | |
|---|---|---|---|---|---|
| | | | Len. | Gap | Time |
| Gurobi Optimization, LLC (2024) | - | - | 1.64 | - | - |
| LKH3 Helsgaun (2017) | - | - | 1.64 | 0.01% | - |
| Nearest Insertion | - | - | 6.60 | 301.65% | - |
| Farthest Insertion | - | - | 6.75 | 310.98 % | - |
| Nearest Neighbor | - | - | 4.69 | 185.26% | - |
| GREAT NB x1 | 1.26M | 10M | 2.00 | 21.53% | 61s |
| GREAT NB x8 | 1.26M | 10M | 1.86 | 13.25% | 9m |
| GREAT NF x1 | 1.26M | 10M | 2.13 | 29.42% | 62s |
| GREAT NF x8 | 1.26M | 10M | 1.98 | 20.64% | 9m |
| GREAT NF x1 | 5.00M | 10M | 2.29 | 39.49% | 2m |
| GREAT NF x8 | 5.00M | 10M | 2.10 | 27.76% | 18m |

## 5 CONCLUSION

In this work, we introduce GREAT, a novel GNN-related neural architecture for edge-based graph problems. While for previous GNN architectures it was necessary to transform graphs into their line graph representation to operate in purely edge-focused settings, GREAT can directly be applied in such contexts. We evaluate GREAT in an edge-classification task to predict optimal TSP edges. In this task, GREAT is able to produce very sparse TSP graphs while deleting relatively few optimal edges compared to heuristic methods. Furthermore, we develop a GREAT-based RL framework to directly solve TSP. Compared to existing frameworks, GREAT offers the advantage of directly operating on the edge distances, overcoming the limitation of previous Transformer and GNN-based models that operate on node coordinates which essentially limits these architectures to Euclidean TSP. This limitation is rather disadvantageous in real-life settings, however, as distances (and especially other characteristics like time and energy consumption) are often asymmetric due to topography (e.g., elevation) or traffic congestion. GREAT achieves promising performance on several TSP variants (Euclidean, asymmetric with triangle inequality, and asymmetric without triangle inequality). We postpone it to future work to adapt GREAT to other routing problems such as CVRP (by translating node demands to edge demands). Furthermore, we aim to develop better data-augmentation methods for GREAT, allowing us to increase optimality at inference time by solving each instance multiple times. We further believe that GREAT could be useful in edge-regression tasks (e.g., in the setting of Hudson et al. (2021)) and, possibly, beyond routing problems.

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

## A    XASY50

Table 5: XASY50 TSP

| Method | Params | Train Set | XASY50 | | |
|---|---|---|---|---|---|
| | | | Len. | Gap | Time |
| Gurobi Optimization, LLC (2024) | - | - | 1.64 | - | - |
| LKH3 Helsgaun (2017) | - | - | 1.64 | 0.0% | - |
| Nearest Insertion | - | - | 4.63 | 181.76% | - |
| Farthest Insertion | - | - | 4.76 | 190.1% | - |
| Nearest Neighbor | - | - | 4.0 | 143.47% | - |
| GREAT NB x1 | 1.26M | 10M | 1.80 | 9.35% | 17s |
| GREAT NB x8 | 1.26M | 10M | 1.73 | 5.48 % | 2m |
| GREAT NF x1 | 1.26M | 10M | 1.88 | 14.74% | 17s |
| GREAT NF x8 | 1.26M | 10M | 1.79 | 9.31% | 2m |
| USL Gaile et al. (2022) | 355K | 12.8M | - | 32.7%* | 9s* |
| SL Gaile et al. (2022) | 355K | 1.28M | - | 83.38%* | 9s* |
| RL Gaile et al. (2022) | 355K | 12.8M | - | 1439.01%* | 9s* |

* evaluated on 1280 instances only

To compare our approach to Gaile et al. (2022), we also train a node-based and node-free GREAT model on this distribution with 50 nodes only and report the results. We report the different results of Gaile et al. (2022) using the different learning paradigms as well as our own results in table 5. Compared to the best result of Gaile et al. (2022), where a GNN-based architecture was trained using USL, our model achieves $3 - 6\times$ better gaps depending on whether we use $\times 8$ instance augmentation or no augmentation at all. We also point out that the RL-based approach of Gaile et al. (2022) was unable to provide meaningful solutions (considering the gap of over $1400\%$) compared to our GREAT model which was also trained using RL.

## B    GREAT-BASED ENCODER-DECODER FRAMEWORK

A visualization for the framework used in the experiments in section 4.2 is shown in fig. 6. For the sake of simplicity, we illustrate the idea of the framework for an Euclidean TSP instance. Non-Euclidean instances can be processed in the same way. The input to the framework is the TSP graph with the corresponding edge weights. GREAT first produces edge encodings from these inputs. In the last GREAT layer, however, the intermediate, internal node encodings of GREAT are returned instead of the edge encodings. This is because the subsequent decoder (adapted from Jin et al.

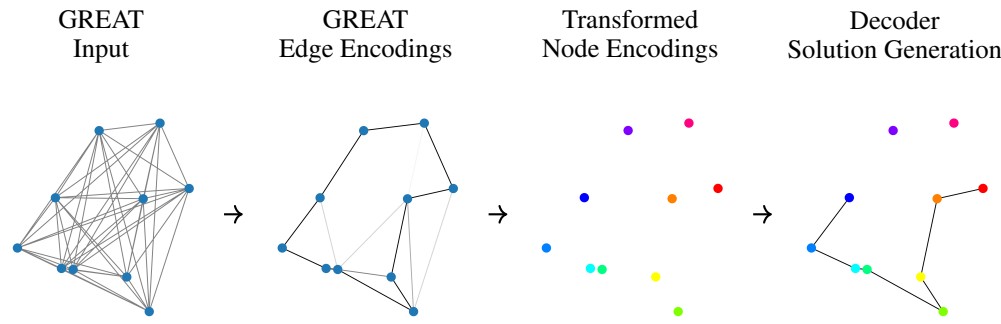

GREAT
Input

GREAT
Edge Encodings

Transformed
Node Encodings

Decoder
Solution Generation

Figure 6: A visualization of our GREAT-based encoder-decoder framework

(2023b)) requires node encodings to iteratively select the next node to add to a partial solution in order to construct a tour.

We visualize in fig. 6 how GREAT potentially learns and passes on information: We imagine that the edge encodings produced by GREAT reflect how promising edges are to be part of the TSP solution. This thought is supported by the fact that GREAT can successfully learn which edges are optimal and which are not as can be seen in section 4.1. In the visualization, we assign darker colors to the encodings of such promising edges. Then, when transformed into node encodings, we assume that the edge information gets passed on and node encodings reflect which nodes are connected by important edges. Our assumption that nodes connected by important edges have similar encodings is backed up by the heatmap visualization in fig. 7. The visualization shows Euclidean distances and cosine similarities between the vector encodings for the nodes in the TSP instance that are returned by the GREAT encoder to be passed on to the decoder. Red frames around a tile in the heatmaps signal that there is an edge between the two nodes in the optimal TSP solution. We can see that for the cosine similarity, the red frames are mostly around tiles with high cosine similarity. Analogously, for the Euclidean distance heatmaps, we can see that the red frames are mostly around tiles with low distances. For generating these heatmaps, we used the node-based GREAT encoder for Euclidean TSP from section 4.2. Even though the model is trained on TSP instances of size 100, we can see that the heatmaps indicate similar patterns for TSP instances of size 50 and 30. In fig. 6, we assign similar colors to nodes that are connected by edges deemed promising by GREAT to visualize the effect shown in the heatmaps. In the last step of our encoder-decoder framework, we hypothesize that the decoder can construct solutions from these embeddings by iteratively selecting similar node encodings.

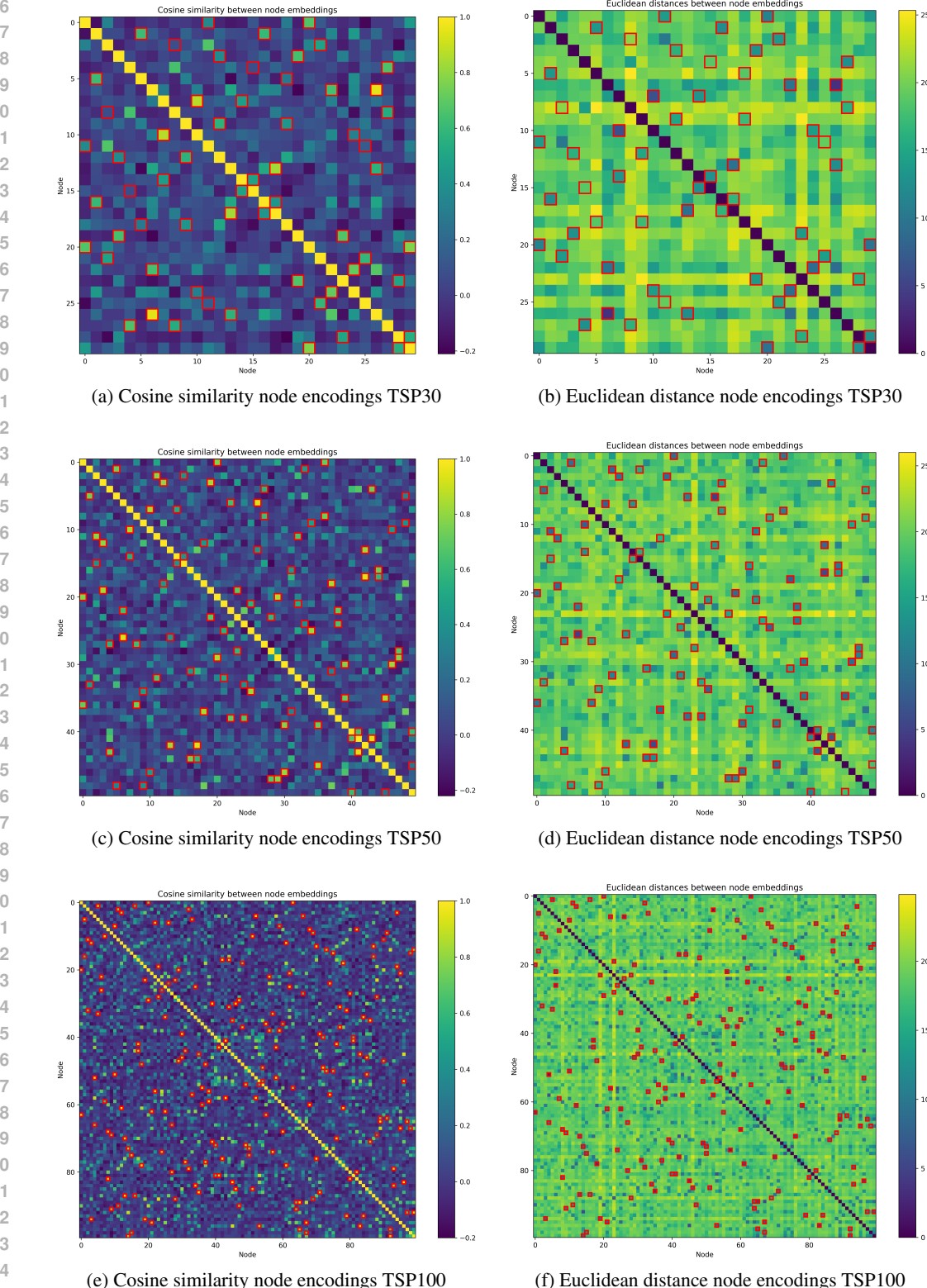

(a) Cosine similarity node encodings TSP30

(b) Euclidean distance node encodings TSP30

(c) Cosine similarity node encodings TSP50

(d) Euclidean distance node encodings TSP50

(e) Cosine similarity node encodings TSP100

(f) Euclidean distance node encodings TSP100

Figure 7: Vector similarities for node encodings returned by the GREAT encoder in the encoder-decoder framework

