# OpenReview forum: "A GREAT Architecture for Edge-Based Graph Problems Like TSP"
_ICLR.cc/2025/Conference — ICLR 2025 Conference Withdrawn Submission_

### Official Review · Reviewer_X3Tu · 2024-10-18

**Soundness:** 2
**Presentation:** 2
**Contribution:** 2
**Rating:** 3
**Confidence:** 4

**Summary:**

The paper introduces Graph Edge Attention Network (GREAT), a new graph neural network architecture designed for edge-based graph problems such as the Traveling Salesman Problem (TSP). The authors evaluate GREAT in two settings: edge classification to predict optimal edges in TSP, and a reinforcement learning framework for constructing TSP solutions. The experiments demonstrate GREAT's ability to produce sparse graphs while retaining most optimal edges and achieve good results on Euclidean and non-Euclidean asymmetric TSP variants.

**Strengths:**

The motivation is well demonstrated and reasonable. GREAT addresses the limitations of traditional GNNs, which are not well-suited for dense graphs and suffer from over-smoothing issues. The model operates on edges rather than nodes, making it ideal for tasks where edge selection is crucial. The authors evaluates GREAT in a supervised setting for edge classification in TSP and a reinforcement learning framework based on GREAT that achieves good performance on asymmetric TSP distributions.

**Weaknesses:**

1. The main experimental evaluation is limited and incomplete. **1) Task level.** If the authors claim GREAT to be a new GNN architecture aimed at edge-based problems, it is expected that the model should be evaluated on more classic edge-level graph tasks such as link prediction and edge classification, similar to what GAT did in its paper. **2) Model level.** If the authors intend to specifically focus on combinatorial optimization (CO) problems on graphs, especially (A)TSP, there are numerous recent methods worth (also conventionally) incorporating for comparison: [1-7] for Euclidean TSP and [8-10] can be applied to ATSP. Given these and the marginal improvements, it is inappropriate to conclude that "on TMAT, the only other available neural solver is MatNet" (line 442), and also, the SOTA claim in the abstract should be reconsidered pending further empirical results. **3) Dataset level.** The evaluated datasets, particularly the one for Euclidean, are not typical. I suggest the authors refer to [1-7] and perform supplementary experiments on the conventionally inherited uniform dataset (1280 instances for TSP-50/100 and 128 instances for TSP-500, and/or TSPLIB, etc), thus enabling consistent comparison with previous works. Note that conducting a subset of the mentioned comparisons is acceptable; the important thing is that they result in a more complete and consistent evaluation as seen in mainstream ML4CO literature.

2. Several counterintuitive results raise questions but are not analyzed thoroughly or convincingly. Further ablation studies are needed to support the effectiveness of GREAT. For example, **1)** experiments comparing your node-free network to previous GNNs/GATs using random, handcrafted, or trainable embeddings as initial node features; **2)** when and why do the node-free and node-based models outperform each other on specific tasks? (lines 437 & 465) **3)** when and why do more or fewer parameters perform better on which tasks? (line 451). These points are mentioned in the paper but lack concrete explanations or additional experiments.

3. The paper structures its content around two main tasks, namely, 1) learning to sparsify and 2) learning to solve non-Euclidean TSP. Regarding the first task, focusing solely on "precision and recall in determining optimal TSP edges" is somewhat ill-suited to evaluate TSP fully. This section could be improved by taking an additional step, i.e., employing post-inference search approaches (greedy, beam search, MCTS, etc.) to convert your "predictions of optimal edges" into actual TSP tours and reporting their optimality gaps with respect to tour length from (near-)optimal solvers. Additionally, since sparsification operations are often performed on larger instances (e.g., $N \geq 500$), the experiments should probably include larger-scaled data to ensure practicality. In summary, I suggest the authors rearrange Section 4.1 and Section 4.2 to discuss TSP/ATSP or SL/RL separately (and also explicitly), which would likely be more reasonable.

4. In the proposed RL framework, **1)** why can't GREAT also serve as the decoder part to form a complete architecture, similar to what MatNet does? **2)** As described in lines 417 to 426, the fact that there is "a final node-based layer where the results of the temporary node features are returned instead of further processing them to obtain edge embeddings again" seems to undermine the significance of your node-free architecture motivation, which is also somewhat counterintuitive.

5. Since the authors emphasize routing problems and more, **1)** can GREAT be applied to a broader range of routing problems (e.g., variants of VRPs, OP, etc.) or other edge-based problems? How capable is the RL framework when generalized to larger instances than seen in the training set, given that most existing RL methods (AM, POMO, SYMNCO, etc.) struggle with scalability, and a toy size of up to 100 nodes is far from sufficient to address real-world CO problems.

6. Some minor concerns: **1)** do the arrows in Fig. 3 & Fig. 4 seem reversed? **2)** How is the "# edges" calculated in Table 1? Is it the total number of edges across 100 instances? If so, you may want to make a note of it. Adding F1-scores in Table 1 would also be beneficial if you insist evaluating TSP solved by your supervised GREAT from the perspective of precision and recall (instead of lengths and gaps of the ultimately decoded tours) of edge selection.

**References:**
[1] DIMES: A Differentiable Meta Solver for Combinatorial Optimization Problems.

[2] An Efficient Graph Convolutional Network Technique for the Travelling Salesman Problem.

[3] T2T: From Distribution Learning in Training to Gradient Search in Testing for Combinatorial Optimization.

[4] Unsupervised Learning for Solving the Travelling Salesman Problem.

[5] DIFUSCO: Graph-based Diffusion Solvers for Combinatorial Optimization.

[6] Graph Neural Network Guided Local Search for the Travelling Salesperson Problem.

[7] Generalize a Small Pre-trained Model to Arbitrarily Large TSP Instances.

[8] GOAL: A Generalist Combinatorial Optimization Agent Learner.

[9] BQ-NCO: Bisimulation Quotienting for Efficient Neural Combinatorial Optimization.

[10] GLOP: Learning Global Partition and Local Construction for Solving Large-scale Routing Problems in Real-time.

**Questions:**

Please refer to the weaknesses part for questions and suggestions.

---

> ### Author Response · Authors · 2024-11-21
>
> Thank you very much for your review and the suggested improvements. We will update the paper and resubmit it at a later point. However, we still want to answer your questions below:
>
> 1)	Thank you for suggesting further experiment settings and baselines to us, we will make use of these suggestions in an updated version of the paper.
>
> 2)	Thank you for suggesting further experiment settings and baselines to us, we will make use of these suggestions in an updated version of the paper.
>
> 3) We do not directly aim to solve TSP in a supervised fashion in the experiments of section 4.1, the task is only optimal vs. non-optimal TSP edge classification.
>
> 4)	By transforming the edge-features into node-features we reduced the amount of embeddings passed to the decoder from n^2 to n. We acknowledge that a different strategy for keeping the edge features would also be possible.
>
> 5)	For GREAT we focused on developing a more neural solver for more realistic non-Euclidean settings of routing problems. Many other works e.g. assume Euclidean TSP and focus on solving larger and larger instances. We, on the other hand, focused on “smaller” instances of only 100 nodes but on a more realistic setting in the sense of asymmetric TSP. We will, however, add experiments for other CO problems in an updated version of the paper.
>
> 6)	Thank you for these suggestions, we will incorporate them in an updated version of the paper.

---

### Official Review · Reviewer_AEKr · 2024-10-22

**Soundness:** 2
**Presentation:** 2
**Contribution:** 2
**Rating:** 3
**Confidence:** 4

**Summary:**

The paper introduces Graph Edge Attention Network (GREAT), a novel neural architecture designed for edge-based graph problems such as the Traveling Salesman Problem (TSP). GREAT operates on edge features directly, unlike traditional Graph Neural Networks (GNNs) that work on node features. The authors evaluate GREAT in two scenarios: predicting optimal edges for TSP in a supervised manner and incrementally constructing TSP solutions within a reinforcement learning framework.

**Strengths:**

There is little works for coordinate-free TSP solver. So the work is a highly original one based on GNN, which is proved to work on the problem.

The proposed GNN, GREAT, is a node-free GNN, different from previous GNNs. Though designed for the coordinate-free TSP, the work might have some inspirations for researchers of GNN from other fields to explore new application scenarios of GNNs.

**Weaknesses:**

1. The model scalability is not demonstrated. Experiments are run on instances of no more than 100 nodes. The methods needs to be run on more than 1000 nodes to demonstrate the scalability.

2. The claim of “state-of-the-art” in the abstract might be over-claiming. Reasons are listed as follows:

a. Some recent works, e.g. GLOP, have also achieved to surpass MatNet on ATSP. Since the comparison with these works are inaccessible, it is very hard to claim that GREAT achieves state-of-the-art.

b. I did not see that GREAT surpasses MatNet in terms of tour length or running time.

c. On the Euclidean 2D data, many available baselines (e.g. DIFUSCO) are missing. So it is very hard to say that GREAT is sota.

d. By the way, we usually say a TSP solver is sota, e.g. [2], when it not only outperforms deep learning-based solver but also outperforms or at least be comparable with heuristics e.g. LKH. Apparently, GREAT has not achieved that yet.

3. The technical contribution might be limited since GREAT focuses on the encoding procedure while adopting the same training pipeline as POMO.

4. Some details of experiments are confusing.
- In Sec. 4.1 authors conduct experiments on sparsification by preserving edges that are more likely to be in the optimal solution. What if the preserved edges do not form a valid tour? How do you deal with this kind of situation?
- In Sec. 4.2, the time used for model training is not reported. I think the training efficiency is also important for comparison between the deep learning methods.
- In Table 3, why don’t you report a length of Matnet before normalization? The results of matnet seem weird, also making reviewers hard to check the correctness.

5. The writing of results analysis in experiments is not clear. Important points are hard to find and follow.

6. As authors said, the covered problems are limited. The proposed GREAT also can be applied to other routing problems e.g. CVRP. Additional experiments on other problems can make the work look more experimentally solid.

7. Though authors promised that the code and other source files would be available after review, they are not given to reviewers to make the results convincing.

Minors:
- Figure 1 is not referred in the main text, where the term “classic GNN” may cause unnecessary misunderstanding that what the exact type of GNN is referred to or it is just a general framework.

- What is the differences between the proposed GREAT and the reference “EdgeFormer” in line 133? This needs to be clarified in the main text.

**Questions:**

See weakness.

---

> ### Author Response · Authors · 2024-11-21
>
> Thank you very much for your review and the suggested improvements. We will update the paper and resubmit it at a later point. However, we still want to answer your questions below:
>
> Weaknesses:
>
> 1)	Our approach works end-to-end (and is also trained this way) and is not devide-and-conquer based. Therefore, training a model for 1000 nodes or more is not easily possibly due to memory requirements and consequent hardware requirements. Compared to other works that often focus on scalability, we focus on generalizing to non-Euclidean, asymmetric instances.
>
> 2)	Thank you for these complementary works, we will include them in an updated version of our paper and adjust our wording regarding the sota-claims.
>
> 3)	It is correct that we follow the existing training pipeline of POMO which has proven itself powerful in the past. However, by adapting the new GREAT-based encoder we are able to generalize to non-Euclidean TSP (and potentially other routing problems).
>
> 4)
>
> a.	We do not directly aim to solve TSP in a supervised fashion in the experiments of section 4.1, the task is only optimal vs. non-optimal TSP edge classification.
>
> b.	It is true that training time can also be interesting, although these times also depend on the used hardware and general code optimization. We will add training times in an updated version of the paper.
>
> c.	The reported values are directly taken from the MatNet-paper by Kwon et al., the reported normalization is different because the authors there normalized the instances differently than us.
>
> 5)	We will try to improve the presentation.
>
> 6)	We will add further experiments on other routing problems in an updated version of the paper.
>
> Minors:
>
> Thank you for the hints, we will adapt them in an updated version of the paper.

---

### Official Review · Reviewer_p9xt · 2024-11-04

**Soundness:** 1
**Presentation:** 1
**Contribution:** 1
**Rating:** 3
**Confidence:** 5

**Summary:**

Summary: The authors proposed an edge-attention-based GNN to solve the TSP problem. Their GNN model aims to tackle the common issues in traditional GNNs, 1) over-smoothing, 2) (they claimed) Besides Euclidean TSP, the model can solve the non-Euclidean variants. They claimed the following contributions: 1) The model can simplify complex instances into "easier" instances by keeping promising edges -- those likely to be considered as part of the optimal solution(s). 2) They build a RL framework capable of solving both Euclidean TSP and its variants.

**Strengths:**

Pros: The topic is machine learning for combinatorial optimization(CO). Unlike general ML use cases, CO is challenging even for modern deep learning models. Investigating challenging use cases like CO is noteworthy to envelope the machine learning techniques.

**Weaknesses:**

Cons: After carefully reading the paper, my concerns are with the technique, the experiments and the presentation. First, I use the following bullet points to summarize my concerns on the technique:

- I understand that the authors intend to emphasize their empirical contributions. However, I didn't see much novelty in the idea of "edge-attention". I understand that it seems reasonable for edge-centric problems like TSP, but my major question is: what distinguishes the proposed model from the normal node-based attention mechanisms, even normal GNNs? Does the attention mechanism do the work, or the concatenation operations do the work (ref. eq. (6) and (7))

- The authors claim that the model can solve Euclidean TSP and some of the variants. Based on my understanding of the model, I believe it is only because the model aggregates the bidirectional messages. For instance, if the problem is asymmetric TSP, then there would be 4 pieces of messages for each edge-attention. If so, I believe this should not be counted as a part of novelty since aggregating necessary information for either edges/nodes is a regular operation for message passing algorithms, especially GNNs.

- The authors mentioned the WL isomorphism testing algorithm several times, but it doesn't directly connect to the major argument of the paper. If I am right, despite the authors clearly mention "In this paper, we overcome the limitations of regular GNNs ....", they neither empirically/theoretically prove that their model has any breakthrough, nor do they run any experiments on the skeptical graphs, i.e. random regular graphs.

Second, I would like to discuss the experiments

- My top concern with the experiments is: the authors have claimed "we overcome the limitations of regular GNNs ....". Then I believe it makes sense to compare the performance with models that use node-based GNNs. For example, [Oriol Vinyals et al. 2015][1], [Marcelo O. R. Prates et al. 2018][2], [Yan Jin et al.][3], [Joshi, Chaitanya K. et al. 2021][4] and even some relatively older version, i.e. [Hanjun Dai et al.][5]. Additionally, there are some similar ideas out there, might also be worthy to make a comparison: [Kun Lei et al.][6](I am not sure about the performance of their model, but this one looks highly related.)

- On the dataset: it is understandable for authors to share the data they used afterwards, but at least they should give a clear description(from a high level) about the scale/magnitude of the datasets. For example, in table 2, 3 and 4, the authors demonstrate the "Gap" in percentile. To me, it's not convincing. For example, if the optimal solution for a TSP instance is 100,000,000, 0.1% is still a huge gap.

- On time: I am not sure if efficiency is a priority for the authors. I believe it should be for any novel solvers. The authors make the comparison using Gurobi as the optimality. Here are my doubts: 1) what are the configurations of Gurobi? For example, which model the authors used in the experiments? Is it a basic one or a sophisticated one like branch-and-cut-and-price? How did the authors tune the parameters of Gurobi? To me, those are very unclear. Even if it were clarified in the paper, I would like to see how well Gurobi can do within 0.01s, 1s and etc. Because for Gurobi, "still running" does NOT mean it hasn't found a good solution.

- On params: why do we need to care how many parameters the model has? If we have to, why does it matter? It influences which aspects? I am unclear on what the authors are trying to demonstrate from this index.

- Additionally, if the authors want to emphasize the performance breakthrough of the proposed model, it is very necessary to compare it with the best solver. If it can't outperform the best one, we should resort to the "latest(the recent year or two years)" neural solvers as mentioned above.

Third, I want to discuss the presentation with the authors. One sentence summary: I think the paper is not very well-written. And I have the following reasons:

1) The paper has many irrelevant content. For example, the authors mentioned WL isomorphism testing several times without any highly relevant usage or proof.

2) Some of the notations are not clearly defined. For example, "NB" and "NF" in Table 2, Table 3 and Table 4. I guess they mean "node-based" and "node-free" respectively.

3) It looks like the abbrev.(FF) for "feedforward" seems unnecessary. "Feedforward" has been mentioned before FF is defined, and it isn't frequently used in the paper. Similar issues can also be found in section 4.1.

4) For Table 2, 3 and 4: What do "k" and "M" mean? I understand that conventionally they represent "a thousand" and "a million", respectively, but the authors should clearly define their usage.

5) The diagrams are not well presented. For example, subscripts for edge notations are exposed as "_".


[1]: https://arxiv.org/abs/1506.03134
[2]: https://doi.org/10.1609/aaai.v33i01.33014731
[3]: https://arxiv.org/abs/2304.09407
[4]: https://arxiv.org/abs/2006.07054
[5]: https://arxiv.org/abs/1704.01665
[6]: https://arxiv.org/abs/2105.02730

**Questions:**

Please refer to the weaknesses section. My questions are clearly stated.

---

> ### Author Response · Authors · 2024-11-21
>
> Thank you very much for your review and the suggested improvements. We will update the paper and resubmit it at a later point. However, we still want to answer your questions below:
>
> Concerns on techniques:
>
> 1)	The proposed model, GREAT, is essentially completely different from other GNNs since it operates on an edge-level compared to node-level. This is visualized in Figure 1 and 2.
>
> 2)	Other GNNs are typically node-based and therefore not designed for routing problems. For Euclidean TSP, this limitation can easily be overcome by using the coordinates of the nodes as node-level inputs. However, this does not work for e.g. asymmetric TSP where there are no coordinates from which the model can internally derive the actual distance metrics. Since GREAT is edge-based, it is directly applicable and, to the best of our knowledge, novel.
>
> 3)	We mention WL since this algorithm “shows” that GNNs are ill-suited for complete (i.e. regular) graphs. We acknowledge that this limitation of GNNs can be overcome by assigning node identifiers like coordinates. However, the issue of over-smoothing in complete graphs remains nevertheless.
>
> Discussion of experiments:
>
> 1)	[Vinyals et al 2015] is not a state-of-the-art paper anymore, [Prates et al 2018] solved the decision variant of TSP, [Jin et al 2023] is quite similar to the AM + POMO by [Kwon et al 2020] which we already include. [Lei et al 2021] is another node-based GNN that does therefore not generalize to non-Euclidean TSP – a limitation we overcome with GREAT.
>
> 2)	The optimal average solution of the data is given in Tables 2,3,4 and 5 as it is usually done in other papers tackling TSP with machine learning.
>
> 3)	Like other papers tackling TSP with machine learning, we provide the results of an optimal solver as a baseline to provide the gaps of our solution. Like other studies using neural models to solve CO problems, we do not aim to perform benchmarks with Gurobi or other solvers. This is also why we (like many other papers) do not provide the runtimes Gurobi achieved on our hardware.
>
> 4)	More parameters mean more training data required (which is often limited in real-world settings). Moreover, we can drastically save on energy and time required at both training and inference time. Furthermore, smaller models can run on cheaper/less potent hardware.
>
> Presentation of the paper:
>
> 1)	Thank you, we will adapt the paper in this regard.
>
> 2)	Thank you for the hint, NB and NF indeed mean node-based and node-free. We will adjust this.
>
> 3)	Thank you for the hint, we will adapt the paper in this regard.
>
> 4)	Thank you for the hint, we will adapt the paper in this regard.
>
> 5)	Thank you for the hint, we will adapt the paper in this regard.

---

### Official Review · Reviewer_MZ6y · 2024-11-12

**Soundness:** 2
**Presentation:** 2
**Contribution:** 1
**Rating:** 3
**Confidence:** 5

**Summary:**

The paper proposes a new neural architecture to tackle graph problems with a focus on edge information. The architecture, called GREAT, consists of a combination of the attention mechanism and message passing along the edges. GREAT is used for two tasks related to solving the traveling salesman problem. The first one is a supervised edge classification task to predict whether or not an edge is part of the optimal solution and serves as a preprocessing step to sparsify the TSP graphs. The second is a constructive approach where a solution is built incrementally by predicting the next node to add to a partial solution. In this case the GREAT model is used within an existing RL-based constructive framework. The approach is evaluated on both euclidian and non-euclidian TSP datasets.

**Strengths:**

* The paper addresses tasks on non-sparse graphs for which the majority of GNNs are not well suited
* The paper is clear and well organized, with nice illustrations in Fig 3 and 4 of the idea of the edge-based message passing

**Weaknesses:**

* There have been many variations of combining GNNs with Transformers, the paper cites some (Sec 2.2) and the proposed architecture ia slight variation of existing ones.
* Missing important references/baselines:
   * [1] solves the same edge classification task with excellent performance on the euclidian TSP. Since this is the same setting as in Sec 4.1, it should be used as a learning-based baseline, as well as the cited [Joshi et al 2019]
   * [2] addresses the asymetric TSP with a combination of transformers and graph convolutional networks, with a superior performance to MatNet and should be used as a baseline in Sec 4.2
   * [3] also proposes an extension of Transformers for graphs with edge features
* The evaluation of the edge classification task by itself brings limited value, I would have liked to see the impact of the proposed architecture on the target TSP task, i.e. using the generated heatmap with some search method to generate TSP solutions, as was done in  [Joshi et al 2019] and [1].
* The statement that the proposed approach achieves state-of-the-art results (L26) is not supported by the experiments:
     * Table 2: "MatNet x8" clearly outperforms the GREAT variants
     * Tables 3 and 4 (asymetric TSP): "GREAT NFx8" performance is in-between MatNet x1 and x128. So no clear gains. It would be interesting to compare to MatNet say x8 (or x16 etc) as in the previous table.
     * The paper claims that no other neural solver has been evaluated on the XASY dataset (L458), but the codes are available so at least MatNet and [2] should be included as baselines.
* No generalization results for TSP instances with more than 100 nodes.
* Finally, although the paper claims that the proposed architecture could be used for other tasks, it is only evaluated on the TSP, which has been extensively studied in the neural combinatorial optimization literature, hence limiting the potential impact of the paper.

[1] Sun et al, DIFUSCO: Graph-based Diffusion Solvers for Combinatorial Optimization, Neurips 2023

[2] Drakulic et al, BQ-NCO: Bisimulation Quotienting for Efficient Neural Combinatorial Optimization, Neurips 2023

[3] Henderson et al, Transformers as Graph-to-Graph Models, ACL 2023

**Questions:**

* L215: the authors claim that node features can easily be transformed into edge features. Can they elaborate on this? This is important since the proposed architecture assumes only that all features are on the edges.
* Regrading the normalization, the paper states L394: "we normalize each instance individually such that the biggest distance is exactly 1" and then L405 "the distances of all instances in the current data batch were multiplied by a factor in the range (0.5, 1.5) to ensure the models learn from a more robust data distribution". Is this done before or after the first normalization?

---

> ### Author Response · Authors · 2024-11-21
>
> Thank you very much for your review and the suggested improvements. We will update the paper and resubmit it at a later point. However,  we still want to answer your questions below:
>
> Answer to weaknesses:
>
> 1)	To the best of our knowledge, we proposed the first purely edge-based GNN. Other works combining GNNs and Transformers operate on a node level (e.g., in routing problems the coordinates of the cities). Some works, including the ones in Sec. 2.2., augment the models to additionally consider edge features when, e.g., computing the attention scores. Nevertheless, these models are node-based unlike GREAT which is purely based on edges.
> The closest other architecture is Edgeformer which was developed for NLP-tasks on textual-edge network data. However, the architecture is still different, since Edgeformer assumes initial node features whose information is injected into the edge features during computation.
>
> 2)
> * [Sun et al 2023] and [Joshi et al 2019] did not perform the same edge classification task as we did, they just used these outputs as components in their overall learning pipelines.
> * Thank you for pointing out [Drakulic et al 2023] to us, we will include this work in an updated version of our paper.
> * Thank you for the hint, this paper is again “node”-based, however, and not edge-based like GREAT.
>
> 3)	Thank you for the input, we did not aim to develop two completely different GREAT-based TSP solvers and decided to go for an end-to-end-based RL approach. We will think about building a second, supervised-learning-based pipeline based on our results in section 4.1.
>
> 4)	We will adjust our wording regarding the sota performance.
>
> 5)	The developed ML-based solver is trained end-to-end in an RL setting. This approach offers the advantage of not needing an additional search algorithm to transform intermediate outputs from an ML model into valid solutions but comes with the limitation of being hardware-extensive during training time which is why it cannot be easily trained for instances with, e.g., 1000 nodes. While it is possible to use GREAT to develop other frameworks that can be used for instances of these sizes, we decided to tackle non-Euclidean and asymmetric instances of smaller sizes instead which is also challenging and highly relevant in real-world scenarios.
>
> 6)	We will add additional experiments for other settings.
>
> Answers to Questions:
>
> 1)	We tried to explain this idea in the paragraph following in 215: The idea is that (for certain settings) node-level features directly translate to edge-level features. Consider CVRP for example: If a customer i has a demand $c_i$, it means that we have to “carry” this demand if we visit this customer. Therefore, each edge $e_ji$ which leads us to this customer has this demand too and we can interpret $c_{i}$ as an edge feature of edge $e_{ji}$.
>
> 2)	"we normalize each instance individually such that the biggest distance is exactly 1" refers to the data-generation or preprocessing step. "the distances of all instances in the current data batch were multiplied by a factor in the range (0.5, 1.5) to ensure the models learn from a more robust data distribution" is done during the actual forward pass of the model. So, to answer the question, this is done after the first normalization.

---

### Note · Authors · 2024-11-21

**Comment:**

We appreciate the reviewers' time and their valuable suggestions for improvement. We will revise the paper accordingly and plan to resubmit it at a later date.

**Withdrawal Confirmation:**

I have read and agree with the venue's withdrawal policy on behalf of myself and my co-authors.